# Activation of VGLL4 Suppresses Cardiomyocyte Maturational Hypertrophic Growth

**DOI:** 10.3390/cells13161342

**Published:** 2024-08-13

**Authors:** Aaron Farley, Yunan Gao, Yan Sun, Sylvia Zohrabian, William T. Pu, Zhiqiang Lin

**Affiliations:** 1Masonic Medical Research Institute, 2150 Bleecker St, Utica, NY 13501, USA; afarley@mmri.edu (A.F.);; 2Department of Cardiology, The Fourth Affiliated Hospital of Harbin Medical University, 37 Yiyuan Street, Harbin 150001, China; 3Department of Cardiology, Boston Children’s Hospital, 300 Longwood Ave, Boston, MA 02115, USAwilliam.pu@cardio.chboston.org (W.T.P.)

**Keywords:** Hippo-YAP pathway, YAP, TEAD, VGLL4, cardiomyocyte, maturational hypertrophy

## Abstract

From birth to adulthood, the mammalian heart grows primarily through increasing cardiomyocyte (CM) size, which is known as maturational hypertrophic growth. The Hippo-YAP signaling pathway is well known for regulating heart development and regeneration, but its roles in CM maturational hypertrophy have not been clearly addressed. Vestigial-like 4 (VGLL4) is a crucial component of the Hippo-YAP pathway, and it functions as a suppressor of YAP/TAZ, the terminal transcriptional effectors of this signaling pathway. To develop an in vitro model for studying CM maturational hypertrophy, we compared the biological effects of T3 (triiodothyronine), Dex (dexamethasone), and T3/Dex in cultured neonatal rat ventricular myocytes (NRVMs). The T3/Dex combination treatment stimulated greater maturational hypertrophy than either the T3 or Dex single treatment. Using T3/Dex treatment of NRVMs as an in vitro model, we found that activation of VGLL4 suppressed CM maturational hypertrophy. In the postnatal heart, activation of VGLL4 suppressed heart growth, impaired heart function, and decreased CM size. On the molecular level, activation of VGLL4 inhibited the PI3K-AKT pathway, and disrupting VGLL4 and TEAD interaction abolished this inhibition. In conclusion, our data suggest that VGLL4 suppresses CM maturational hypertrophy by inhibiting the YAP/TAZ-TEAD complex and its downstream activation of the PI3K-AKT pathway.

## 1. Introduction

From birth to adulthood, the heart size increases around 20 times, which is mainly driven by CM maturational hypertrophic growth [1]. Besides increasing their size, during the maturation of a neonatal heart to an adult heart, mammalian CMs undergo a remarkable transformation from proliferative to post-mitotic with several specializations simultaneously laid out, which make them uniquely suited for sustaining billions of forceful, rhythmic contractions. The mechanisms that regulate and implement maturation are being robustly investigated due to the recent advance in induced pluripotent stem cell (iPSC)-based cardiac regenerative medicine because a major barrier hampering the progress of this field is the limited maturity of CMs that can be generated from iPSC. One of the fruitful strategies to understand CM maturation is to study the molecular signaling pathways controlling the development of naturally occurring mammalian CMs, such as neonatal rodent CMs, and the new knowledge originating from these studies will facilitate the effort of generating iPSC-derived mature CMs. 

As one of the maturational adaptations, maturational hypertrophy is regulated by both growth hormones and mechanical forces [2]. For example, glucocorticoid and thyroid hormones play essential roles in CM maturational hypertrophic growth. In mice, the disruption of glucocorticoid signaling leads to myofibril disorganization and heart dysfunction [3]. In sheep, elevated thyroid hormone level decreases CM proliferation and increases CM size [4]. Recently, Parikh and his colleagues reported that combinational treatment of T3 (triiodothyronine) and Dex (dexamethasone, a synthetic glucocorticoid) promoted the hypertrophy of human iPSC-derived CMs. This work also showed that T3/Dex induction of iPSC-CM T-tubule formation relied on the presence of patterned Matrigel mattress, thus providing direct evidence that growth hormones and mechanical stress synergize to regulate CM maturational hypertrophy [2]. 

The Hippo-YAP pathway is a key pathway that regulates heart development and regeneration [5]. The canonical Hippo-YAP pathway is composed of a kinase cascade that conveys the cell contact signals from cell membrane to the terminal transcriptional effectors, most notably YAP (Yes-associated protein) and its homolog WWTR1 (WW domain containing transcription regulator 1, also known as TAZ). Activated YAP and TAZ interact with a number of transcription factors, such as TEAD1, to potently stimulate proliferation and promote cell survival [6]. Vestigial-like 4 (VGLL4) is another crucial terminal effector of the Hippo-YAP pathway, which competes against YAP/TAZ for TEAD binding [7]. We previously showed that overexpression of VGLL4 in the neonatal hearts caused cardiac hypoplasia [8] and cardiac-specific depletion of VGLL4 did not affect postnatal heart growth [9]. Nevertheless, it remains unclear whether activation of VGLL4 interferes with CM maturational hypertrophic growth. 

The phosphoinositide 3-kinase (PI3K)-AKT pathway is one of the downstream effector pathways regulated by the YAP/TAZ-TEAD complex [10,11], and PI3K has been known for its crucial role in controlling CM maturational hypertrophic growth: under baseline conditions, genetic activation of PI3K increased heart size, and inhibition of PI3K decreased heart size, and neither of these two genetic manipulations of PI3K affected heart function [12]. 

In the current work, we established T3/Dex induction of neonatal rat ventricular myocytes (NRVMs) hypertrophy as an in vitro CM maturational hypertrophy model and investigated the role of VGLL4 in CM maturational hypertrophy. We demonstrated that the activation of VGLL4 suppressed CM maturational hypertrophy both in vitro and in vivo. On the molecular level, the activation of VGLL4 suppressed the PI3K-AKT pathway, which could not be recapitulated by a mutated VGLL4 that minimally interacts with TEAD1. Additionally, we found that VGLL4 promoted the expression of Cathepsin B (CTSB), which in turn caused TEAD1 degradation. Together, our data suggest that the activation of VGLL4 suppresses CM maturational hypertrophic growth and that VGLL4 functions by inactivating the YAP/TAZ-TEAD complex and the downstream PI3K-AKT pathway. 

## 2. Materials and Methods 

### 2.1. Experimental Animals

Wild-type Swiss Webster (CFW) mice were used in this study. CFW mice were originally purchased from Taconic Biosciences, Germantown, NY, USA, and maintained in the AAALAC accredited (#001865) mouse facility of Masonic Medical Research Institute (MMRI). Wistar rat pups were purchased from Charles River Laboratory (Wilmington, MA, USA).

### 2.2. Adenoviruses

Ad-CMV-LacZ adenovirus was purchased from SignaGen Laboratories, Frederick, MD, USA (Catalog #: SL100714). VGLL4-GFP adenovirus was generated by cloning human VGLL4 cDNA with a C-terminal GFP into pENTRY (Invitrogen, Waltham, MA, USA). The H212A/F213A mutations and H240A/F241A mutations were introduced into the human VGLL4 coding sequence by replacing CACTTT with GCAGCT and CATTTC with GCAGCC, respectively. The resulting construct was named pENTRY.VGLL4^HF4A^ GFP. These expression cassettes were then transferred to pAd/CMV/V5-DEST using LR clonase (Invitrogen). For packaging adenovirus, pAd.CMV.VGLL4 and pAd.CMV.VGLL4^HF4A^ were linearized with PacI and transfected into 293Ad cells. Viruses were purified with Adenovirus Purification kit (Biomiga, San Diego, CA, USA) and tittered using the AdEasy adenoviral titer kit (Stratagene, San Diego, CA, USA).

### 2.3. Neonatal Rat Ventricular Myocytes (NRVMs)

NRVMs were isolated from 2-day-old Wistar rats (Charles River) using the Neomyts CM dissociation kit (Cellutron, Nottingham, MD, USA, NC6031). Isolated NRVMs were initially cultured for 24 h in the presence of 10% fetal bovine serum (NS medium, Cellutron, Nottingham, MD, USA, m8031). NRVMs were then cultured in NW (Cellutron, Nottingham, MD, USA, m-8032) medium containing 5% horse serum and 20 μM cytosine B-D-arabinofuranoside (Ara-C) (Sigma, St. Louis, MO, USA, C1768) for 48 h. After washing with phosphate-buffered saline (PBS), the medium was changed into serum free NW medium. One day after serum starvation, NRVMs were applied for different treatments. To test the maturational hypertrophic effects of different hormone stimuli, NRVMs were treated with 100 nM T3, 1 µM Dex, or 100 nM T3 + 1 µM Dex (T/D). To test the hypertrophic effects of YAP/TAZ and VGLL4, NRVMs were infected with adenovirus at a dose of 25 multiplicity of infection (MOI), supplemented with 100 nM T3 + 1 µM Dex (T/D). Cells were collected for analysis 40–48 h post-treatment. 

### 2.4. AAV Transduction and Echocardiography Measurement

Low doses (0.33 × 10^10^ GC/g) of AAV9.cTnT.Luciferase or AAV9.cTnT.VGLL4^K225R^ were subcutaneously injected into wild-type neonatal CFW pups at postnatal day 3 (P3), respectively. Body weight and heart function were measured at P8 and P19. To measure heart function, M-mode echocardiography measurements were performed on conscience mice with a VisualSonics Vevo 3100 (FUJIFILM Sonosite, Toronto, ON, Canada) ultrasound imaging system.

### 2.5. Seahorse XF Mito Stress Test 

NRVMs were seeded in a 96-well Seahorse XF plate. Then, 40–48 h after virus and chemical treatment, components of the Cell Mito Stress Test (Agilent, Santa Clara, CA, USA) were used to evaluate NRVMs mitochondrial function at the following final concentrations: 1.25 µM Oligomycin, 3 mM FCCP, and 1 µM/1 µM rotenone/antimycin A. After analysis, cells were fixed and stained for DAPI and counted for cell number. Cell number was used to normalize readings from the Seahorse XF Analyzer (Agilent, Santa Clara, CA, USA) [13].

### 2.6. Statistics

Normally distributed data values, such as mRNA and protein levels, were expressed as mean ± SD and analyzed with Student’s *t*-test (two groups) or one-way ANOVA followed by Tukey’s post hoc test (more than two groups). Non-parametric data, such as cell size and myofibril thickness/length, were analyzed using Mann–Whitney test (two groups) or Kruskall–Wallis test followed by Dunn’s multiple comparisons (more than two groups). Prism10 (Version 10.1.0) software was used to plot the bar/violin graphs and to perform the statistical analysis. 

## 3. Results

### 3.1. The Expression of Cardiac Lipid Metabolic Genes Increases with Age

As a relatively new topic, CM maturational hypertrophy has not been intensely studied, and the establishment of an easy and reliable in vitro CM maturational hypertrophy model will benefit this field. After birth, CM maturational hypertrophic growth [14] is accompanied with a rapid metabolic shift from glycolysis to fatty acid oxidation [15]. We expected that the expression of lipid metabolic genes should increase as the heart undergoes maturational growth and therefore could be used as molecular markers to reflect CM maturation status. To test our hypothesis, we compared the expression of seven well-established lipid metabolism genes between fetal, neonatal, and adult hearts: Lipoprotein Lipase (*Lpl*), Patatin-Like phospholipase domain containing 2 (*Pnpla2*, also known as *Atgl*), Fatty Acid Binding Protein 3 (*Fabp3*), *Fabp4*, *Cd36*, Carnitine palmitoyltransferase 1A (*Cpt1a*), and Solute carrier family 27 member 1 (*Slc27a1*, also known as *Fatp1*). By combining the published RNA seq data [16] and our unpublished postnatal day 1 heart RNA seq data, we generated a gene expression heat map, which demonstrated that all of these selected lipid metabolism genes were low in the E12.5 fetal heart and high in the postnatal or adult heart (Figure 1A). We then collected hearts from different age mice and performed qRT-PCR to further profile the expression pattern of these genes. Compared to the E18.5 hearts, the neonatal (P5), juvenile (P14), and adult (P42) hearts had higher expression of *Cd36, Fabp3, Fabp4,* and *Cpt1a* (Figure 1B). Between postnatal and adult hearts, the expression of *Cd36* and *Cpt1a* was not distinguishable, and the expression of *Fabp3/Fabp4* was higher in the juvenile heart than in the adult heart (Figure 1B). The expression of cardiac *Slc27a1* and *Pnpla2* showed a tight positive correlation with mouse age (Figure 1C); the expression of *Lpl* did not differ between fetal and postnatal/juvenile heart and robustly increased in the adult heart (Figure 1D). These data suggest that *Slc27a1* and *Pnpla2* are promising molecular markers that can be used to reflect the CM maturation status.

### 3.2. T3/Dex Treatment Promotes CM Maturational Hypertrophy

The combination treatment of T3 and Dex (T/D) is known to promote CM maturation in the presence of a patterned Matrigel mattress [2]; however, it is unclear whether T/D treatment promotes CM maturational hypertrophy in a 2D culture system. Compared with the 3D culture system that relies on the presence of delicate and expensive patterned Matrigel, the 2D culture system is easier to handle and can generate sufficient materials for biochemistry studies at a lower cost. To define which hormone treatment would better recapitulate the CM maturational hypertrophic growth in a 2D culture system, we treated the NRVMs with T3, Dex, and T/D. NRVMs treated with DMSO were used as a control. Compared with the control CMs, all three hormone-treated groups had an enlarged cell size, and the T/D treatment group showed the highest cell size increase (Figure 2A,B); the CM length was increased in the T3 and T/D group (Figure 2C); the CM width was increased in the Dex and T/D groups (Figure 2D); and the length to width ratio was increased only in the T3 group (Figure 2E). The 2D cultured CMs are usually spindle- or triangle-shaped; however, we observed that the Dex-treated CMs grew more protrusions, which made them display a spider-like shape (Figure 2A). This observation was further corroborated by quantifying the protrusion numbers per CM (Figure 2F), which was assessed by counting the number of skinny protrusions (Figure 2A, yellow arrowheads) that originated from the cell body and were independent of the major CM branches (Figure 2A, white arrows). 

Sarcomeres are the basic units of myofibrils that generate contracting force in CMs, and sarcomere maturation is a crucial step for CM maturation [17]. Localized in the thin filament of the sarcomere, the troponin complex regulates the interaction between sarcomere actin and myosin heavy chain heads [18]. To determine to which extent T3 and Dex affect sarcomere maturation, we used the cardiac troponin I (TNNI3) [19] antibody to visualize the sarcomere/myofibril structure in NRVMs. Compared with control NRVMs, T3-and T/D-treated cells had much more organized myofibrils, and Dex treatment did not obviously change the myofibril organization (Appendix A). We then quantified the sarcomere length, myofibril thickness, and length to gain a detailed understanding of how these two hormones regulate myofibrillogenesis. During human CMs maturation, the sarcomere length increases from 1.7 µm to approximately 2.2 µm [17]. We observed that neither T3 nor Dex treatment increased sarcomere length and that T/D treatment slightly decreased sarcomere length (Appendix A). During CM maturation, new sarcomeres are longitudinally and laterally added into the pre-existed myofibrils to increase the length and width of the myofibrils, respectively. Our data showed that T/D treatment increased myofibril width and length, T3 treatment only increased myofibril length, and Dex treatment did not have an obvious effect (Appendix A). During CM maturation, immature slow skeletal slow skeletal troponin I is gradually replaced by mature cardiac troponin I [20]. We then assessed TNNI3 expression levels by measuring its fluorescence intensity. The TNNI3 fluorescence intensity per cell was similar between the control and Dex-treated NRVMs. T3- and T/D-treated NRVMs showed much stronger TNNI3 fluorescence intensity than the control and Dex-treated CMs (Appendix A). These data together suggest that the combination use of T3 and Dex hormones regulates myofibrillogenesis by incorporating more sarcomeres into the pre-existing myofibrils. 

Myosin heavy chain 6 *(Myh6)* and *Myh7* are two cardiac muscle genes abundantly expressed in adult and fetal murine CMs, respectively [21]. In the adult human heart, the dominant myosin heavy chain isoform is MYH7 [22]. All three treatments increased *Myh6* and decreased *Myh7*, whereas T/D had a stronger effect on *Myh6* expression than the other two treatments (Figure 2G). To further define the molecular effects of T3, Dex, and T/D treatment, we examined the expression of the aforementioned seven lipid metabolic genes. The qRT-PCR results showed that T/D treatment increased the expression of *Fabp3, Fabp4, Scl27a1, Pnpla2, and Lpl* (Figure 2H; Appendix A), and the expression of these five genes was not consistently up-regulated by either T3 or Dex (Appendix A). The expression of *Cd36* and *Cpt1a* was not affected by these three hormone treatments (Appendix A). Among these seven lipid genes, *Pnpla2* was robustly increased by T3 and T/D (Figure 2H), and ATGL, the protein product of *Pnpla2*, showed a similar expression pattern as the *Pnpla2* gene (Figure 2I). Thyroid hormone is critical for CM maturation, as indicated by the observation that genetically inactivating thyroid hormone signaling in the CMs delayed CM cell cycle exit and decreased the expression of oxidative phosphorylation genes [23]. We further corroborated our in vitro data by examining the expression of ATGL in the *Myh6-Cre;Thra^DN^* mouse hearts [23] and found that blocking thyroid hormone signaling in the CMs greatly decreased the ATGL protein level (Appendix A). 

Although T/D combination treatment has been shown to promote CM T-tubule formation [24], it is not clear whether they also boost CM metabolic maturation. To address this issue, we treated NRVMs with T/D and assessed mitochondrial function. The Seahorse XF mitochondria stress test is a standard assay to examine mitochondrial respiration activity, in which Oligomycin, Carbonyl cyanide-p-trifluoromethoxyphenylhydrazone (FCCP), and rotenone/antimycin (R/A) are used to inhibit ATP synthase, to disrupt the proton gradient, and to shut down mitochondria respiration, respectively. Using this assay, we measured four parameters related to mitochondrial function: the basal oxygen consumption rate (OCR), proton leak OCR (during Oligomycin treatment), maximal OCR (during FCCP treatment), and non-mitochondrial OCR (during R/A treatment). Compared with the vehicle control, T3 treatment of NRVMs increased all the first three parameters (Figure 2J,K), and Dex treatment alone did not affect mitochondria respiration. Although T/D treatment increased maximal OCR, it had a weaker effect than T3 (Figure 2K). 

Together, these data showed that T/D treatment of NRVMs increased cell size, promoted the expression of adult CM genes, and nurtured oxidative phosphorylation metabolism, thus suggesting that T/D treatment of NRVMs in 2D culture system is a reliable model for studying CM maturational hypertrophy. 

### 3.3. Reducing YAP/TEAD Complex Activity Attenuates CM Maturational Hypertrophy

Although the roles of YAP/TAZ and TEAD1 in CM proliferation and pathological hypertrophy have been well documented [25,26,27], it is not clear whether these factors are associated with CM maturational hypertrophy. To answer this question, we first measured the expression of *Yap* and *Tead1* mRNA in control and T/D-treated NRVMs. Compared with control, T/D-treated NRVMs had significantly higher levels of *Tead1*. Nevertheless, the expression of *Yap* was not affected by T/D treatment (Appendix A). We further performed YAP immunofluorescence staining to examine whether T/D changed YAP cellular distribution and found no difference between control and T/D-treated CMs (Figure 3A). 

In the heart, YAP and TAZ have functional redundancy, as evidenced by the observation that loss of both proteins, but neither alone, caused perinatal death [28]. We previously showed that the loss of YAP did not affect CM maturational hypertrophic growth [1]. We reasoned that the presence of TAZ-TEAD complex may compensate for YAP’s function to support hypertrophic growth in the YAP-null CMs. To circumvent the YAP/TAZ redundancy issue, we chose to use a previously published YAP-TEAD interference peptide (YTIP) [1] to probe the function of the YAP/TAZ-TEAD complex in T/D-treated CMs, expecting that YTIP would block the interaction between YAP/TAZ and the TEAD family proteins (TEAD1,2,3,4). Three treatments were included in this study: Adenovirus LacZ (Ad.LacZ), Ad.LacZ+T/D, and Ad.YTIP +T/D. Compared to the LacZ group, LacZ+T/D treatment increased CM size, and replacing LacZ with YTIP attenuated the hypertrophic effects of T/D (Figure 3B,C). The PI3K-AKT pathway is essential for CM hypertrophic growth [29], and *Pik3ca* and *Pik3cb* are regulated by YAP [10]. We therefore examined the expression of these two genes to validate the YTIP inactivation of the YAP/TAZ-TEAD complex. Our results showed that both *Pik3ca* and *Pik3cb* were increased by T/D and that YTIP decreased the expression of these two genes in the presence of T/D stimulus (Figure 3D).

Together, these data suggest that the YAP/TAZ-TEAD complex is involved in the regulation of CM maturational hypertrophy. 

### 3.4. Activation of VGLL4 Suppresses CM Maturational Hypertrophy In Vitro

In mice, the prenatal heart grows mostly by increasing CM number through proliferation. Shortly after birth, most CMs exit cell cycle [30], and therefore, the heart grows mainly through maturational hypertrophy from birth to adulthood. The cardiac expression of VGLL4 increases with age [8], suggesting that VGLL4 may regulate CM maturational hypertrophic growth. To test this hypothesis, we first investigated VGLL4 function in our newly developed CM maturational hypertrophy model. We generated an adenovirus expressing VGLL4-GFP merge protein (Ad.VGLL4) (Appendix A) and transduced cultured NRVMs in the presence of T/D. Ad. LacZ was used as virus control. T/D treatment increased CM size, and Ad.VGLL4 significantly suppressed T/D-induced CM hypertrophy (Figure 4A,B). Immunofluorescence imaging of the NRVMs revealed that VGLL4 was localized in the nucleus (Figure 4A), and Western blot further confirmed the expression of exogenous VGLL4-GFP in the Ad.VGLL4 transduced CMs (Figure 4C). Compared with the LacZ+T/D group, VGLL4+T/D NRVMs had similar *Myh6* expression levels and lower *Myh7* (Figure 4D), suggesting that VGLL4 differentially regulates the expression of these two muscle genes. 

### 3.5. VGLL4 Suppresses CM Maturational Hypertrophic Growth In Vivo

We previously showed that the acetylation of VGLL4 decreased its interaction with TEAD1 and that the VGLL4^K225R^ mutant had a stronger interaction affinity with TEAD than the wild-type VGLL4 [8]. In the same study, we used a high dose of cardiac-specific adenovirus-associated virus (AAV) to overexpress wild-type VGLL4 (AAV.cTnT.VGLL4) and VGLL4^K225R^ (AAV.cTnT.VGLL4^K225R^) in the neonatal heart, respectively. In line with the biochemistry findings, AAV.cTnT.VGLL4 did not affect postnatal heart growth, while AAV.cTnT.VGLL4^K225R^ (AAV.VGLL4^K225R^) caused heart failure. On the cellular level, AAV.VGLL4^K225R^ induced CM necrosis and increased CM size. These published data suggest that high-dose AAV.VGLL4^K225R^ greatly reduces YAP/TAZ-TEAD complex activity, thus leading to heart failure and secondary CM pathological hypertrophy. 

In the cultured NRVMs, we showed that VGLL4 suppressed CM maturational hypertrophy. To corroborate these in vitro observations and to avoid heart failure-associated secondary effects, we decided to use a low-dose (0.33 × 10^10^ GC/g) AAV.VGLL4^K225R^ to activate VGLL4 in the neonatal heart, so that VGLL4 regulation of CM maturational hypertrophic growth would not be confounded by heart failure that occurred in the high-dose AAV.VGLL4^K225R^ transduced mouse pups [8]. P3 pups were then transduced with a low-dose AAV.VGLL4^K225R^, and their heart size/heart function were checked at P8 and P19. Mouse pups receiving the same dose of AAV9.cTnT.Luciferase (AAV.Luci) were used as control. 

Low-dosage AAV.VGLL4^K225R^ did not affect heart function at P8 but mildly decreased systolic heart function at P19 (Figure 5A). Compared with the AAV9.Luci control, AAV.VGLL4^K225R^ mice had a smaller heart size and heart weight at both P8 and P19 (Figure 5B,C). The body weight of the two groups of mice were similar at P8, and AAV.VGLL4^K225R^ mice displayed a lower body weight at P19 (Figure 5D). Correspondingly, the heart-to-body weight ratio of AAV.VGLL4^K225R^ mice was significantly less at P8 but was not distinguishable from that of the AAV.Luci mice at P19 (Figure 5E). We then checked the CM size at P19 and found that AAV.VGLL4^K225R^ significantly reduced CM size (Figure 5F,G). These data suggest that activation of VGLL4 suppresses CM maturational hypertrophic growth in vivo. 

### 3.6. Activation of VGLL4 Suppresses CM Hypertrophic Growth through TEAD

Because VGLL4 disrupts the YAP/TAZ-TEAD complex by competing against YAP/TAZ for TEAD binding [8,31], we tested whether the inhibition of CM maturational hypertrophy requires VGLL4 binding to TEAD. VGLL4 has two conserved tandem Tondu (TDU) domains mediating its interaction with TEAD, and VGLL4 containing H212A/F213A/H240A/F241A (VGLL4^HF4A^) mutations minimally interacts with TEAD (Figure 6A) [31]. To further investigate the molecular mechanism of VGLL4 regulating CM maturational hypertrophy, we first compared the biochemistry activity of VGLL4 and VGLL4^HF4A^ in our system. In a co-immunoprecipitation assay, the data showed that TEAD1 successfully pulled down VGLL4 but not VGLL4^HF4A^ (Figure 6B). We further used TEAD luciferase reporter [32] to test whether VGLL4^HF4A^ would affect YAP/TEAD transcriptional activity and found that VGLL4 and not VGLL4^HF4A^ robustly suppressed YAP transcriptional activity (Figure 6C). 

We generated an adenovirus expressing VGLL4^HF4A^ (Ad.VGLL4^HF4A^) (Appendix A), with which we would determine whether VGLL4 relied on TEAD to suppress CM maturational hypertrophy. NRVMs were treated with Ad.LacZ, T/D+Ad.LacZ, T/D+Ad.VGLL4-GFP, and T/D+Ad.VGLL4^HF4A^, respectively. Compared to the LacZ control, VGLL4 and not VGLL4^HF4A^ significantly diminished T/D induction of CM hypertrophy (Figure 6,E). qRT-PCR confirmed the overexpression of VGLL4 in the Ad. VGLL4 or Ad. Ad.VGLL4^HF4A^-transduced NRVMs (Figure 6F). In the presence of T3/Dex, neither VGLL4 nor Ad.VGLL4^HF4A^ significantly affected the expression of *Myh6* (Figure 6G); however, VGLL4 robustly decreased the expression of *Myh7* (Figure 6G). In comparison with VGLL4, VGLL4^HF4A^ had a similar suppressive effect on the expression of *Myh7* (Figure 6G). 

In our T/D-NRVMs model, we showed that CM maturational hypertrophic growth was accompanied by an increase of oxidative respiration (Figure 2). Because VGLL4 suppressed T/D-induced CM maturational hypertrophy, we wondered whether VGLL4 also affected CM metabolism maturation. The Seahorse XF mitochondria stress test was then used to address this issue. Compared with the LacZ control, T/D treatment of NRVMs increased the oxidative respiration rate (Figure 6H,I). For the T/D-treated cells, VGLL4 did not affect the baseline OCR but increased the proton leak and maximal OCR; VGLL4^HF4A^ did not affect the baseline and proton leak OCR and increased maximal OCR (Figure 6H,I). Overall, both VGLL4 and VGLL4^HF4A^ increased the CM respiration rate in the presence of T/D, and VGLL4^HF4A^ did not differ from VGLL4 in respect of regulating CM metabolism.

### 3.7. VGLL4 Suppresses PI3K-AKT Pathway

To determine whether VGLL4 suppressed CM hypertrophy by inhibiting the PI3K-AKT pathway, we first measured the expression of *Pik3ca* and *Pik3cb*. qRT-PCR showed that T/D treatment increased the expression of these two genes, which was attenuated by VGLL4 and not VGLL4^HF4A^ (Figure 7A). Because PI3K relays its signals through AKT [33], we then checked whether VGLL4 reduced AKT activity. Compared to control NRVMs, T/D-treated cells had a higher level of phosphorylated AKT and a similar level of total AKT (Figure 7B–D). In the presence of T/D, VGLL4 and not VGLL4^HF4A^ significantly attenuated the level of phosphorylated AKT (Figure 7B,D). These data suggest that activation of VGLL4 suppressed the PI3K-AKT pathway through engaging TEAD. 

### 3.8. VGLL4 Induces TEAD1 Degradation through CTSB

Our previous work suggests that VGLL4 regulates YAP/TAZ-TEAD complex through two different mechanisms: (1) VGLL4 binds TEAD and prevents the formation of YAP/TAZ-TEAD complex; (2) VGLL4 promotes TEAD degradation through a proteasome-independent pathway [8]. In the same work, we found that E64, a specific inhibitor of cysteine proteases Cathepsins B, H, and L [34], attenuated VGLL4-induced TEAD1 degradation. Supporting our findings, Cathepsins B (CTSB) was unbiasedly identified in the Co-IP pulldown product of TEAD2 in a high-throughput proteomics study [35]. We therefore focused on testing whether CTSB was responsible for TEAD1 degradation. 

CA-074 is a selective inhibitor of CTSB [36], and the addition of CA-074 to 293T cells increased the protein level of TEAD1 in the presence of overexpressed VGLL4 (Figure 8A,B). Conversely, overexpressing CTSB robustly decreased the TEAD1 protein level (Figure 8C). Further, the TEAD1 co-immunoprecipitation assay showed that CTSB directly interacted with TEAD1 (Figure 8D). These data suggest that CTSB is a primary cysteine protease responsible for TEAD1 degradation. To explore the possible link between VGLL4 and CTSB, we examined the expression of endogenous CTSB in the NRVMs and found that the overexpression of VGLL4 increased the mRNA and protein levels of CTSB (Appendix A and Figure 8E). Meanwhile, VGLL4 decreased the TEAD1 protein level (Figure 8E). 

In summary, our data suggest that VGLL4 regulates CM maturation through multiple mechanisms. On the one hand, VGLL4 prevents the formation of the YAP/TAZ-TEAD complex by both reducing TEAD’s availability to YAP/TAZ and by promoting TEAD protein degradation, thereby suppressing CM proliferation and hypertrophic growth (anabolic growth). On the other hand, VGLL4 favors the CM to adopt a catabolic metabolism program through an uncharacterized mechanism that is independent of the YAP/TAZ-TEAD complex (Figure 8F).

## 4. Discussion

Induced pluripotent stem cell (iPSC)-based regenerative strategies are being developed to treat heart diseases, and a major barrier to heart regeneration is the limited maturity of CMs that can be generated from iPSC. Overcoming this barrier requires understanding the mechanisms that control the normal maturation of CMs as they transit from neonatal cells to terminally differentiated adult cells. A major hallmark of CM maturation is their increase in size, referred to as maturational hypertrophy [14], and therefore investigating the signaling pathways that control CM maturational hypertrophic growth will provide novel insight for understanding the molecular mechanisms of CM maturation. In this study, we characterized an in vitro 2D cell culture model to study CM maturational hypertrophy and showed that the activation of VGLL4 suppressed CM maturational hypertrophy both in vitro and in vivo. Our results suggest that VGLL4, YAP/TAZ, and TEAD proteins orchestrate CM maturational hypertrophic growth, probably through modulating the activity of the PI3K-Akt pathway. 

T3/Dex combination treatment induction of CM maturational hypertrophy. From the neonatal period to adulthood, mammalian CMs undergo a remarkable maturation process that prepares them with many adaptions to cope with high pumping force. Among these adaptations are (1) maturational hypertrophy; (2) cell cycle exit; (3) metabolic switch from glycolysis to mitochondrial oxidative phosphorylation, with concomitant expansion and morphological changes in mitochondria; (4) structural modifications such as T-tubules that enhance excitation–contraction coupling; and (5) altered gene expression, including changes in sarcomere isoforms and expression of the cardioprotective gene [14]. Although important, little is known about the molecular mechanisms that dictate the dramatic cellular structure and gene expression changes during CM maturation. In this scenario, developing an in vivo CM maturational hypertrophy model will greatly benefit CM maturation research. Glucocorticoid and thyroid hormones are both critical for CM maturation, and the combination use of T3 and Dex has been reported to promote the maturation of iPS-CMs in a 3D culture system [24]. In this study, we compared the biological effects of T3, Dex, and T3/Dex in cultured NRVMs, and our data showed that T3/Dex combination treatment is superior to either T3 or Dex single treatment in these examined parameters: cell shape, cell size, and maturation gene expression. Additionally, T3/Dex treatment increased the CM oxidative respiration rate. Our OCR data suggest that Dex diminishes T3′s metabolic effects. Since Dex may counteract T3 action either through enhancing T3 to metabolic inactive reverse T3 (rT3) conversion [37] or by decreasing thyroid hormone receptor expression [38], we reasoned that T3′s strong metabolic effects can be dampened by Dex so that the T3/Dex-treated CMs have a better-balanced metabolism program than the T3-treated CMs. These data together suggest that T3/Dex-treated NRVMs can be used to study CM maturational hypertrophy. 

Pnpla2 as a molecular marker of CM maturation. Fatty acid supplements have been shown to promote iPS-CM maturation, such as maturational hypertrophy and an increase in contraction force [39], highlighting the importance of lipid metabolism in CM maturation. In this study, we profiled the expression of seven lipid metabolism genes in mouse hearts and in cultured NRVMs, with four of them responsible for fatty acid transport (*Slc37a*, *Cd36*, *Fabp3*, and *Fabp4*) and two of them in charge of lipolysis (*Lpl* and *Pnpla2*). Among these seven genes, *Pnpla2* stood out as a gene that is tightly regulated by mouse age. Additionally, *Pnpla2* and its protein product were synergistically regulated by T3 and Dex. The protein product of *Pnpla2* is a triglyceride lipase (ATGL) that catalyzes the key-step biochemical reaction of breaking down triglycerides into glycerol and free fatty acids. In mice, the loss of ATGL causes cardiac steatosis and blunts the function of PPAR-α and PGC-1, two master regulators of energy homeostasis and lipid metabolism, thus leading to severe impairment of mitochondrial respiration and heart function [40]. Based on the literature and our newly acquired data, we suggest that, besides the well-established Myh6 and Myh7 muscle genes mirroring CM maturation status, *Pnpla2* can be used as another reliable molecular marker to monitor CM maturation.

The YAP/TAZ-TEAD complex is responsible for CM maturational hypertrophic growth. It has been well documented that the Hippo-YAP pathway regulates embryonic heart development and cardiac regeneration [5], but the roles of this pathway in CM maturational hypertrophy have not been sufficiently addressed. We previously showed that the deletion of YAP did not affect CM hypertrophic growth in the postnatal heart [1]; however, because YAP and TAZ have functional redundancy in the heart, it is not clear whether TAZ compensates YAP’s function in the YAP-null CMs. As a strong suppressor of YAP/TAZ, VGLL4 prohibits the formation of YAP/TAZ-TEAD complex by engaging TEAD [31]. Therefore, VGLL4 serves as a natural tool to probe the function of the YAP/TAZ-TEAD complex during postnatal heart development. Our current data show that activation of VGLL4 in the postnatal heart suppresses heart growth and decreases CM size. In vitro mechanisms studies demonstrate that VGLL4’s suppression of CM hypertrophy depends on its interaction with TEAD. This new strain of evidence supports the hypothesis that the Hippo-YAP pathway regulates CM maturational hypertrophy. 

VGLL4 regulates CM maturational hypertrophy. In our previous study, we showed that high-dose AAV.VGLL4^K225R^-mediated precautious activating VGLL4 in the neonatal heart caused heart failure, and the CMs of these failure hearts underwent pathological hypertrophic growth [8]. In this study, to test whether VGLL4 regulates CM maturational hypertrophy, we used a low-dose AAV.VGLL4^K225R^ (one third of the published high dose) to transduce the neonatal mouse hearts. In these low-dose AAV.VGLL4^K225R^-transduced mice, their hearts were smaller than the AAV.Luci-transduced littermate controls, and their CM size was also reduced. Consistent with these in vivo data, VGLL4 also suppressed T3/Dex-induced maturational hypertrophy in vitro. Different from the gain-of-function studies, cardiac-specific knocking out VGLL4 did not affect postnatal heart growth [9]. We reasoned that the other three homologs of VGLL4 (VGLL1, VGLL2, and VGLL3) [41] might compensate for VGLL4′s function in the VGLL4 conditional knockout mice. Together, the published work and the newly obtained data suggest that VGLL4 is sufficient but not necessary for suppressing CM maturational hypertrophy. 

By manipulating the expression of VGLL4 in cultured CMs, we find that maturational hypertrophic growth can be separated from metabolism maturation because overexpressing VGLL4 attenuated T3/Dex-induced hypertrophy and simultaneously increased the CM oxidative respiration rate. Further molecular mechanism studies lead us to conclude that VGLL4 suppresses CM maturational hypertrophy by inhibiting the YAP/TAZ-TEAD complex and that VGLL4 regulates CMs metabolism maturation independent of TEAD. A similar observation that maturational hypertrophy and metabolism maturation are two independent processes has been described in one of our previous works, which shows that CM-specific depletion of *Tfam*, the master regulator of mitochondria biogenesis, impairs CM respiration without affecting CM maturational hypertrophic growth [42]. Based on the current data and the literature, we propose that CM maturation is controlled by signaling networks composed of multiple signaling pathways and that VGLL4 serves as a node that connects the CM hypertrophic growth and metabolism maturation signaling pathways.

Limitations of the current study. One of the intriguing findings of this work is that VGLL4 regulates CM metabolism independent of TEAD. Because our current work aims to define the function of VGLL4 in the regulation of CM maturational hypertrophy, we only performed mitochondria respiration tests and did not investigate the molecular mechanisms of how VGLL4 regulates CM metabolism maturation. This is out of the focus of the current study and will be an interesting topic to follow.

## 5. Conclusions

The combined use of T3 and Dex partially improves neonatal rat cardiomyocyte maturation in a 2D culture system, including maturational hypertrophic growth, myofibril formation, and metabolism maturation. The YAP/TAZ-TEAD complex is required for supporting T3/Dex-induced cardiomyocyte maturational hypertrophic growth, and VGLL4 suppresses this process without affecting cardiomyocyte metabolism maturation. 

## Figures and Tables

**Figure 1 cells-13-01342-f001:**
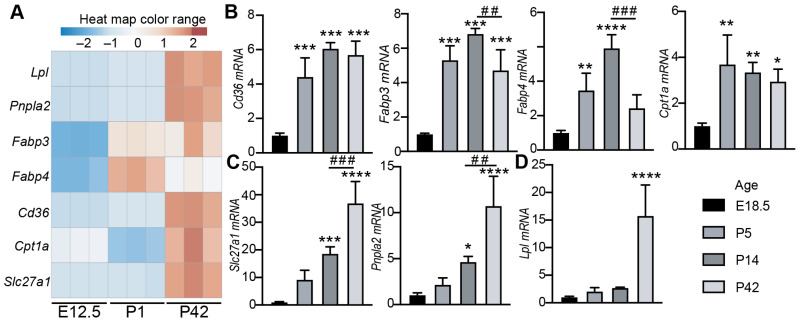
The expression of cardiac lipid metabolism genes increases with age. (**A**). Heat map illustrating the differential expression data of selected cardiac lipid metabolism genes. E12, embryo age 12.5 days; P1, postnatal day 1; P42, postnatal day 42. (**B**–**D**). qRT-PCR measurement of cardiac lipid metabolism genes. RNA isolated from the hearts of different age mice were tested. For each group, four hearts were included. Statistical analysis was performed with one-way ANOVA followed by Tukey’s multiple comparison test. * indicates comparisons between E18.5 hearts and the other age hearts. *, *p* < 0.05; **, *p* < 0.01; ***, *p* < 0.001; ****, *p* < 0.0001. # indicates comparisons between P14 and P42 hearts. ##, *p* < 0.01; ###, *p* < 0.001.

**Figure 2 cells-13-01342-f002:**
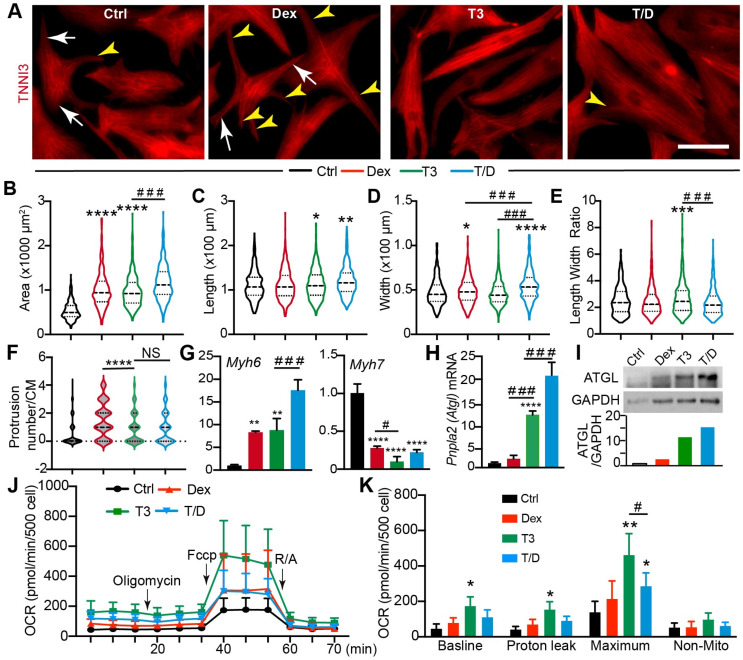
T3/Dex treatment promotes CM maturational hypertrophic growth. (**A**) Immunofluorescence images of NRVMs. Scale bar = 50 µm. White arrows indicate slim cell body; yellow arrowheads indicate protrusions extending out of the cell bodies. (**B**–**E**) Quantification of cell size (**B**), length (**C**), width (**D**), and length-to-width ratio (**E**). (**F**) Quantification of protrusion number per CM. Violin plot was used to display the distribution of CM populations that were categorized by protrusion number per cell. NS, not significant. (**B**–**F**) statistical analysis was performed with Kruskal–Wallis ranks test followed by Dunn’s multiple comparison test. * indicates comparisons between control and the other groups. * *p* < 0.05; **, *p* < 0.01; ***, *p* < 0.001; ****, *p* < 0.0001. # indicates comparisons between indicated groups. ###, *p* < 0.001. For each group, 400–450 CMs were analyzed. (**G**,**H**) qRT-PCR measurement of NRVMs gene expression levels. Statistical analysis was performed with one-way ANOVA followed by Tukey’s multiple comparison test. * indicates comparisons between control and the other groups. **, *p* < 0.01; ****, *p* < 0.0001. # indicates comparisons between indicated groups. #, *p* < 0.05; ###, *p* < 0.001. For each group, *n* = 4. (**I**) Western blot of ATGL. Proteins isolated from NRVMs were analyzed. GAPDH was used as loading control. In the bar graph, ATGL protein level was normalized to GAPDH. (**J**,**K**) Mitochondria respiration activity measured by Seahorse XF. A total of 500 NRVMS were seeded and treated with indicated hormones for 2 days before being subjected to mitochondria stress test. OCR: oxygen consumption rate. (**J**) real-time measurement of OCR. (**K**) quantification of OCR under baseline and different stress conditions. Statistical analysis was performed with one-way ANOVA followed by Tukey’s multiple comparison test. * indicates comparisons between control and the other groups. *, *p* < 0.05; **, *p* < 0.01. # indicates comparisons between indicated groups. #, *p* < 0.05. For each group, *n* = 6.

**Figure 3 cells-13-01342-f003:**
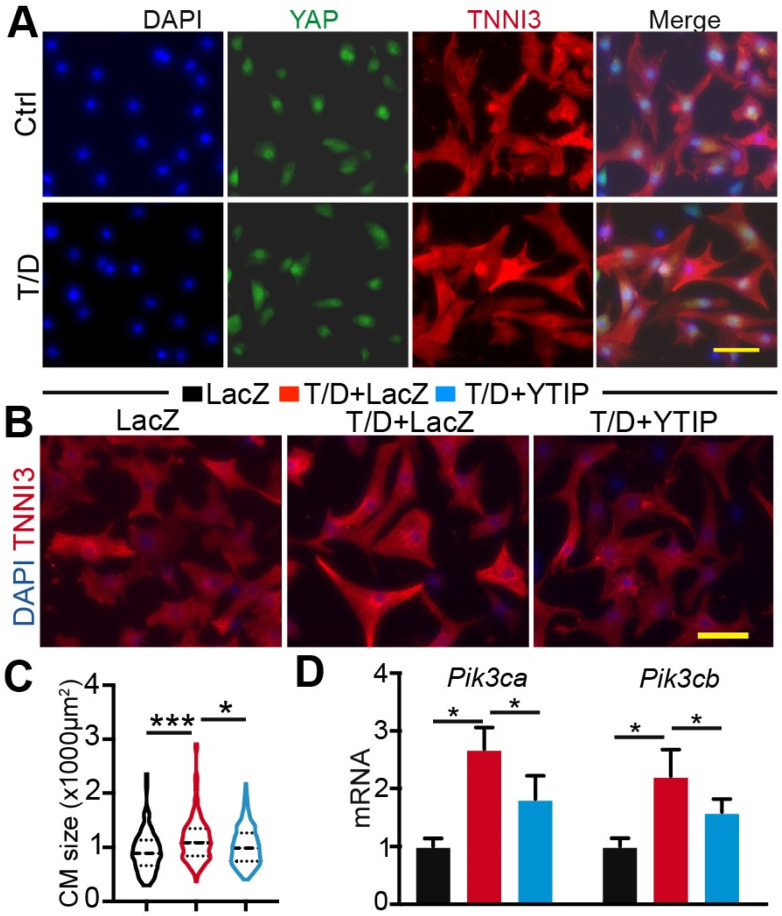
T3/Dex induction of CM maturational hypertrophy depends on YAP/TAZ-TEAD complex. (**A**) YAP immunofluorescence staining. (**B**) Representative immunofluorescence images of NRVMs. (**A**,**B**) scale bar: 50 µm. (**C**) Quantification of NRVMs surface area. For each group, 150–200 cells from 4 biological replicates were measured. Kruskal–Wallis ranks test followed by Dunn’s multiple comparison test, *, *p* < 0.05; ***, *p* < 0.001. (**D**) qRT-PCR measurement of gene expression. 2 days after T3/Dex and YTIP treatment, NRVMs were collected for gene expression analysis. One-way ANOVA test followed by Tukey’s multiple comparison test: *, *p* < 0.05; *n* = 4.

**Figure 4 cells-13-01342-f004:**
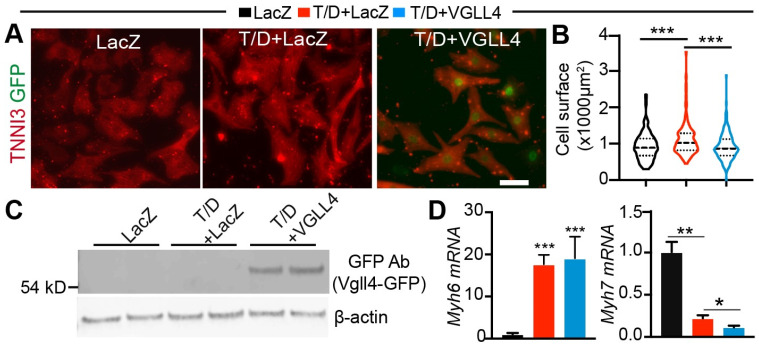
Activation of VGLL4 suppresses T3/Dex-induced hypertrophic growth. NRVMs were treated with 100 nM T3 + 1 µM Dex (T/D) for 40–48 h to induce hypertrophy. Ad.VGLL4-GFP was used to activate VGLL4. (**A**) NRVMs stained with TNNI3 antibody. Scale bar: 50 µm. (**B**) Measurement of NRVMs surface area. A total of 200–350 cells from 4 biological replicates were measured. Kruskal–Wallis ranks test followed by Dunn’s multiple comparison test: ***, *p* < 0.001. (**C**) Western blot of NRVMs transduced with Ad.VGLL4-GFP. β-actin was used as loading control. (**D**) qRT-PCR measurement of *Myh6* and *Myh7*. One-way ANOVA followed by Tukey’s multiple comparison test: *, *p* < 0.05; **, *p* < 0.01; ***, *p* < 0.001; *n* = 4.

**Figure 5 cells-13-01342-f005:**
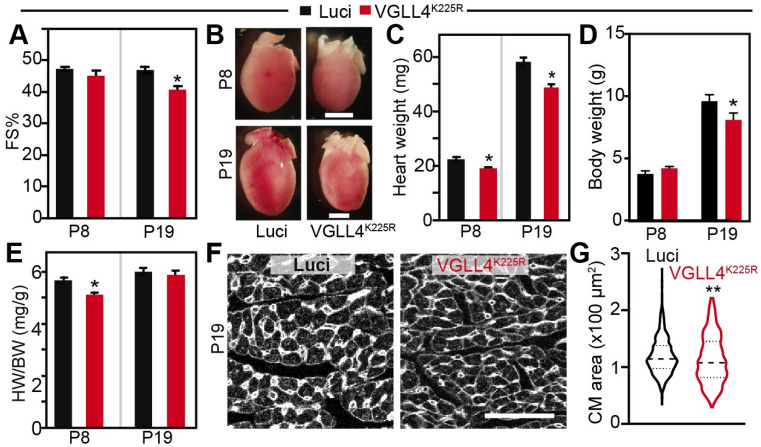
Activation of VGLL4 suppresses CM maturational hypertrophic growth in vivo. AAV9.VGLL4^K225R^ was subcutaneously injected into 3-day-old mouse pups. AAV9.Luci-transduced mouse pups was used as control. (**A**) Fraction shortening (FS%) measured by echocardiography at postnatal day 8 (P8) and day 19 (P19). Measurements were carried out on conscious mice. (**B**) Gross morphology of hearts. Scale bar: 2 mm. (**C**) Heart weight. (**D**) Body weight. (**E**) Heart-to-body weight ratio. (**A**,**C**–**E**) for each group, 4 mice were included. Student *t*-test, * *p* < 0.05. (**F**) Representative images of heart sections stained with Wheat Germ Agglutinin (WGA). Scale bar: 50 µm. (**G**) CM cross-section area. A total of 4 hearts from each group were used for quantification. A total of 654 and 862 CMs were measured in Luci and VGLL4^K225R^ hearts, respectively. Mann–Whitney test: **, *p* < 0.01.

**Figure 6 cells-13-01342-f006:**
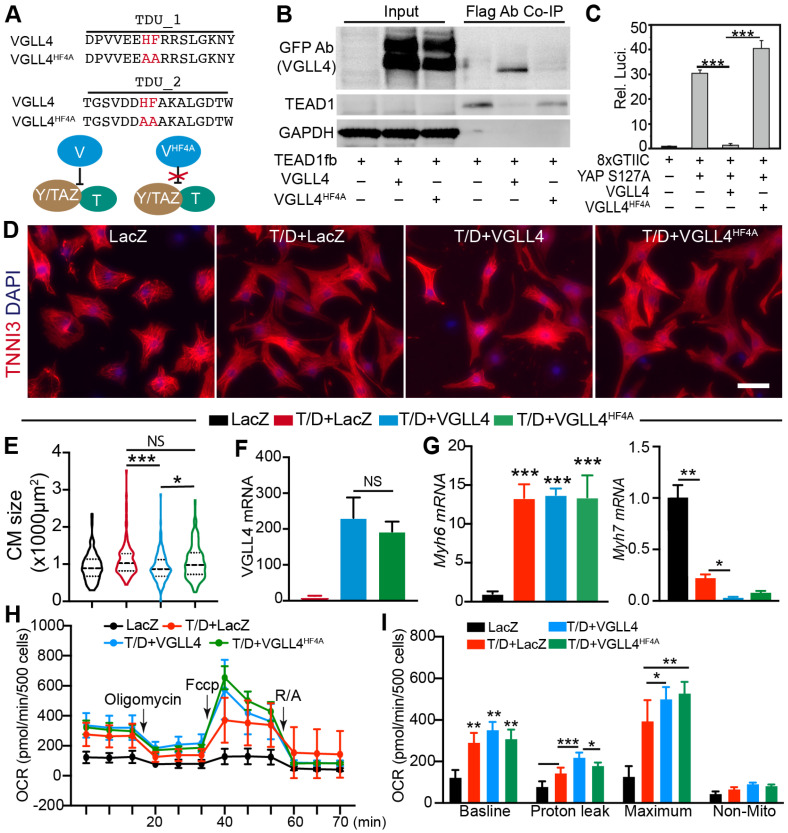
VGLL4 suppresses T3/Dex-induced CM hypertrophy through TEAD. (**A**) Schematic view of the TDU domain sequences of wild-type and HF4A-mutated human VGLL4. Wild-type VGLL4 (V) and not the HF4A-mutated VGLL4 (V^HF4A^) blocks the formation of YAP/TAZ (Y/TAZ)–TEAD (T) complex. (**B**) Co-immunoprecipitation (Co-IP) assay. Indicated plasmids were transfected into 293T cells. 24 h post transfection, cells were lysed and applied for Co-IP. (**C**) Dual luciferase reporter assay in 293T cells. Dual luciferase assay was performed 24 h after transfection. 8xGTIIC: TEAD luciferase reporter plasmid. N = 6. (**D**) Representative immunofluorescence images of NRVMs. Scale bar: 50 µm. (**E**). Quantification of NRVMs surface area. For each group, 200–300 cells from 4 biological replicates were measured. Kruskal–Wallis ranks test followed by Dunn’s multiple comparison test: *, *p* < 0.05; *** *p* < 0.001. NS, not significant. (**F**,**G**) qRT-PCR measurements. 2 days after T3/Dex and indicated adenovirus treatment, NRVMs were collected for gene expression analysis. One-way ANOVA followed by Tukey’s multiple comparison test: *, *p* < 0.05; **, *p* < 0.01; ***, *p* < 0.001, *n* = 5. NS, not significant. (**H**,**I**) Mitochondria respiration activity measured by Seahorse XF. (**H**) real-time measurement of OCR. (**I**) quantification of OCR under baseline and different stress conditions. Statistical analysis was performed with one-way ANOVA followed by Tukey’s multiple comparison test. * *p* < 0.05; ** *p* < 0.01, ***, *p* < 0.001; *n* = 6.

**Figure 7 cells-13-01342-f007:**
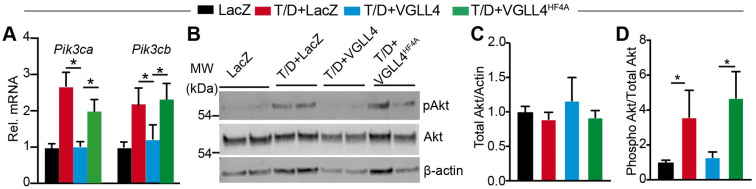
VGLL4 suppresses PI3K-Akt pathway. (**A**) qRT-PCR measurement of *Pik3ca* and *Pik3cb* expression levels. N = 4. (**B**) Western blot of phospho Akt (Ser 473) and total Akt. β-actin was used as loading control. (**C**) Densitometric quantification of total Akt. The total Akt densitometric value was normalized to that of β-actin. (**D**) Ratio of phospho Akt and total Akt. One-way ANOVA followed by Tukey’s multiple comparison test, *, *p* < 0.05; *n* = 3.

**Figure 8 cells-13-01342-f008:**
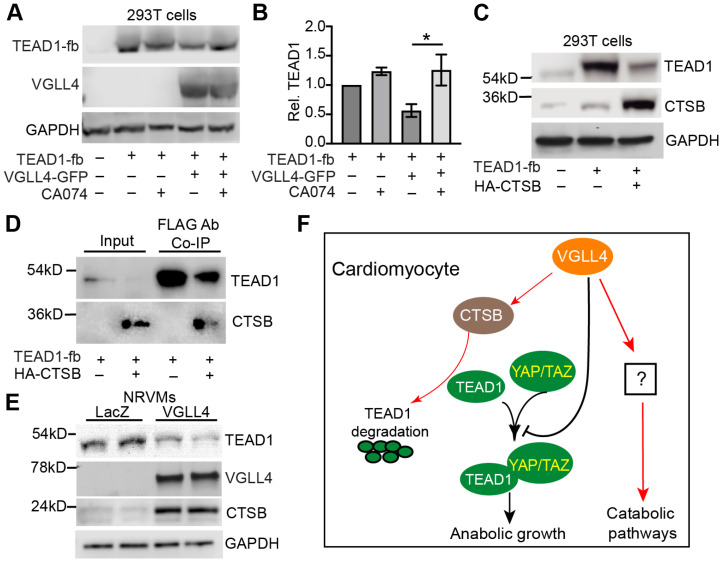
VGLL4 promotes TEAD1 degradation through CTSB. (**A**) Western blot of TEAD1 and VGLL4. 293T cells were transfected with indicated plasmids. 24 h after transfection, cells were treated with 10 µM CA074 for 12 h before being collected for Western blot analysis. GAPDH was used as loading control. (**B**) Densitometric quantification of TEAD1. TEAD1 densitometric value was normalized to that of GAPDH. One-way ANOVA followed by Tukey’s multiple comparison test, *, *p* < 0.05. N = 3. (**C**) Western blot of TEAD1 and CTSB. 293T cells were transfected with indicated plasmids. 24 h after transfection, cells were collected for Western blot analysis. (**D**) Co-immunoprecipitation (Co-IP) assay. Indicated plasmids were transfected into 293T cells. 24 h post-transfection, cells were lysed and applied for Flag Ab Co-IP. (**E**) NRVMs Western blots. Serum-starved NRVMs were treated with indicated virus for 36–40 h before being collected for Western blot analysis. GAPDH was used as loading control. (**F**) Summary of the current study.

## Data Availability

The original contributions presented in the study are included in the article/Appendix A; further inquiries can be directed to the corresponding author.

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
