# Peer review of "Activation of VGLL4 Suppresses Cardiomyocyte Maturational Hypertrophic Growth"

_cells, 2024, doi:10.3390/cells13161342_

Round 1

Reviewer 1 Report

Comments and Suggestions for Authors

Transition from hyperplasia to hypertrophic growth is a key step of postnatal heart development. The current manuscript by Farley et al described a role of VGLL4 in postnatal hypertrophic growth of cardiomyocytes. The authors established a useful in vitro model for studying hormone-induced cardiomyocyte growth. The manuscript is well written. Some minor concerns will need to be addressed.

Concern:

1. Figure 2, does the T/D treatment improve the maturation of sarcomeric structure of NRVMs? 

2. In the NRVMs, does T/D change Yap subcellular localization? or affect its phosphorylation rate?

3. Figure 5, did activation of VGLL4 affect the non-cardiomyocyte population? In panel F, it seems that the coronary endothelial cell population is altered by the VGLL4 activation. Also, it would be more appropriate to dissociate the cardiomyocytes for size measurement. Please also check the labeling of this figure, I think the name of the VGLL4 mutant is not correct.

4. It is interesting that iCM maturation by T3/Dex needs patterned Matrigel mattress, but the NRVMs don't. Are there any explanation? If Matrigel provide "hint" of biomechanics, maybe the NRVMs receive such "hint" before they were dissociated. Please discuss.

Author Response

Reviewer 1: 

Transition from hyperplasia to hypertrophic growth is a key step of postnatal heart development. The current manuscript by Farley et al described a role of VGLL4 in postnatal hypertrophic growth of cardiomyocytes. The authors established a useful in vitro model for studying hormone-induced cardiomyocyte growth. The manuscript is well written. Some minor concerns will need to be addressed.

We thank the reviewer for the critical comments, and we have tried our best to address the raised concerns. In the following paragraphs are the point to point responses.

Concern:

1. Figure 2, does the T/D treatment improve the maturation of sarcomeric structure of NRVMs? 

Response: We examined the sarcomere/myofibril structures, and fount that T/D treatment increased the myofibril width and length. The data were presented in the new Supplemental Fig 2. We also added one paragraph in the text to describe our findings. 

Lines: 168 to 191

"Sarcomeres are the basic units of myofibrils that generate contracting force in CMs, and sarcomere maturation is a crucial step for CM maturation [17]. Localized in the thin filament of the sarcomere, troponin complex regulates the interaction between sarcomere actin and myosin heavy chain heads [18]. To determine to which extent T3 and Dex affect sarcomere maturation, we used cardiac troponin I (TNNI3)[19] antibody to visualize the sarcomere/myofibril structure in NRVMs. Compared with controls NRVMs, T3 and T3/D treated cells had much more organized myofibrils; however, Dex treatment did not obviously change myofibril organization (Suppl. Fig. 2A). We then quantified the sarcomere length, myofibril thickness and length, to have a detail understanding how these two hormones regulate myofibrillogenesis. During human CMs maturation, the sarcomere length increases from 1.7 µm to approximately 2.2 µm [17]. We observed that neither T3 nor Dex treatment increased sarcomere length, and that T3/D treatment slightly decreased sarcomere length (Suppl. Fig. 2B). During CM maturation, new sarcomeres are longitudinally and laterally added into the pre-existed myofibrils, to increase the length and width of the myofibrils, respectively. Our data showed that T3/D treatment increased myofibril width and length, and T3 treatment only increased myofibril length; however, Dex treatment did not have an obvious effect (Suppl. Fig. 2C and 2D). During CM maturation, immature slow skeletal slow skeletal troponin I is gradually replaced by mature cardiac troponin I [20]. We then assessed TNNI3 expression levels by measuring its fluorescence intensity. The TNNI3 fluorescence intensity per cell was similar between control and Dex treated NRVMs, and T3 or T3/D treated NRVMs showed much stronger TNNI3 fluorescence intensity than the control and Dex treated CMs (Suppl. Fig. 2E). These data together suggest that the combination use of T3 and Dex hormones regulates myofibrillogenesis by incorporating more sarcomeres into the preexisting myofibrils" 

2. In the NRVMs, does T/D change Yap subcellular localization? or affect its phosphorylation rate?

Response: As shown in Figure 3A, we found that T/D treatment did not change YAP sub-cellular localization, suggesting that YAP phosphorylation rate is not affected by T/D. Additionally, T/D treatment increased Tead1 and not Yap expression (Suppl. Fig.3). These data suggest that T/D treatment mainly affects YAP/TEAD1 complex by regulating TEAD1. 

3. Figure 5, did activation of VGLL4 affect the non-cardiomyocyte population? In panel F, it seems that the coronary endothelial cell population is altered by the VGLL4 activation. 

Response: The CMs plasma membrane was stained by WGA, which also non-specifically labels endothelial cells. We examined more heart section images and did not find a consistent endothelial density/distribution pattern. With limited data, we are not sure whether activating VGLL4 in CMs affect non-CM population. Although this is an interesting question, it is out of the scope of the current study and therefore not being further investigated. 

Also, it would be more appropriate to dissociate the cardiomyocytes for size measurement. 

Response: During preparing the revision, we had technical difficulties of dissociating the P19 hearts, mainly because the aorta was too small and fragile. At last we managed to dissociate one VGLL4K225R and one GFP control heart when these mice reached to one-month of age. The results showed a trend of VGLL4K225R decreasing the CM width; however,  the results were not conclusive ( one heart/ group) and therefore were not presented. 

Please also check the labeling of this figure, I think the name of the VGLL4 mutant is not correct.

Response: the name of the mutant was corrected. 

4. It is interesting that iCM maturation by T3/Dex needs patterned Matrigel mattress, but the NRVMs don't. Are there any explanation? If Matrigel provide "hint" of biomechanics, maybe the NRVMs receive such "hint" before they were dissociated. Please discuss.

Response: CM maturation spans a spectrum of physiological and structural changes. As shown in our current work, T3/Dex promotes NRVMs  maturational hypertrophy and increases NRVMs fatty acid oxidation activity; however, short period of T3/Dex treatment does not promote the formation of rod-shape CMs. Therefore, T3/Dex treatment of 2D cultured CMs promotes their maturation to some extent, but does not generate fully mature CMs. 

Matrigel provides a 3D environment that CMs can take advantage to form muscle bundles, which is a crucial step for the young CMs to become mature rod-shape adult CMs. We reasoned that CM maturation is a complicated process driven by both intrinsic (e.g., T3/Dex) signaling and environmental factors (e.g, spatial dimension and mechanical stress), and that only providing one category of factors is not enough to generate fully mature CMs. 

Reviewer 2 Report

Comments and Suggestions for Authors

The study by Aaron Farley et al. discusses the role of the Hippo-YAP signaling pathway in maturational hypertrophy. They investigated Vestigial like 4 (VGLL4) as a crucial component of the Hippo-YAP pathway and found that it suppresses CM maturational hypertrophy by inhibiting the YAP/TAZ-TEAD complex and its downstream activation of the PI3K-AKT pathway. This paper addresses an interesting topic. However, the reviewer has the following concerns before it can be accepted:

  1. The condition of the sarcomere is also crucial for evaluating the maturation of cardiac cells. Have the authors evaluated related factors such as sarcomere length and related markers?
  2.  
  3. In the maturation process, the MYH6 expression level typically decreases while the MYH7 level increases. However, in the present study, the addition of T/D decreased the MYH7 level while increasing the MYH6 level. Please comment on this difference.
  4.  
  5. The TNNI3 level is also an important maturation marker, which could be up-regulated during the maturation process. The reviewer is curious whether there is a change in the present study. Please analyze the expression level (immunostaining) in Figure 2A and Figure 3A.
  6.  
  7. In the statistical method section, please specify which data belong to normally distributed data and which are non-parametric data.
  8.  
  9. Please specify the full names of FCCP and R/A in the OCR data figure legend for better reader understanding. Additionally, the OCR data show that T/D has a weaker effect than T3 alone. Please discuss the possible reasons for this.

Author Response

Reviewer 2: 

The study by Aaron Farley et al. discusses the role of the Hippo-YAP signaling pathway in maturational hypertrophy. They investigated Vestigial like 4 (VGLL4) as a crucial component of the Hippo-YAP pathway and found that it suppresses CM maturational hypertrophy by inhibiting the YAP/TAZ-TEAD complex and its downstream activation of the PI3K-AKT pathway. This paper addresses an interesting topic. However, the reviewer has the following concerns before it can be accepted:

We thank the reviewer for the critical comments, and we have tried our best to address the raised concerns. In the following paragraphs are the point to point responses.

The condition of the sarcomere is also crucial for evaluating the maturation of cardiac cells. Have the authors evaluated related factors such as sarcomere length and related markers?

Response: We examined the sarcomere/myofibril structures, and fount that T/D treatment increased the myofibril width and length. The data were presented in Supplemental Figure 2. We also added one paragraph in the text to describe our findings. 

Lines: 168 to 191

"Sarcomeres are the basic units of myofibrils that generate contracting force in CMs, and sarcomere maturation is a crucial step for CM maturation [17]. Localized in the thin filament of the sarcomere, troponin complex regulates the interaction between sarcomere actin and myosin heavy chain heads [18]. To determine to which extent T3 and Dex affect sarcomere maturation, we used cardiac troponin I (TNNI3)[19] antibody to visualize the sarcomere/myofibril structure in NRVMs. Compared with controls NRVMs, T3 and T3/D treated cells had much more organized myofibrils; however, Dex treatment did not obviously change myofibril organization (Suppl. Fig. 2A). We then quantified the sarcomere length, myofibril thickness and length, to have a detail understanding how these two hormones regulate myofibrillogenesis. During human CMs maturation, the sarcomere length increases from 1.7 µm to approximately 2.2 µm [17]. We observed that neither T3 nor Dex treatment increased sarcomere length, and that T3/D treatment slightly decreased sarcomere length (Suppl. Fig. 2B). During CM maturation, new sarcomeres are longitudinally and laterally added into the pre-existed myofibrils, to increase the length and width of the myofibrils, respectively. Our data showed that T3/D treatment increased myofibril width and length, and T3 treatment only increased myofibril length; however, Dex treatment did not have an obvious effect (Suppl. Fig. 2C and 2D). During CM maturation, immature slow skeletal slow skeletal troponin I is gradually replaced by mature cardiac troponin I [20]. We then assessed TNNI3 expression levels by measuring its fluorescence intensity. The TNNI3 fluorescence intensity per cell was similar between control and Dex treated NRVMs, and T3 or T3/D treated NRVMs showed much stronger TNNI3 fluorescence intensity than the control and Dex treated CMs (Suppl. Fig. 2E). These data together suggest that the combination use of T3 and Dex hormones regulates myofibrillogenesis by incorporating more sarcomeres into the preexisting myofibrils"  

In the maturation process, the MYH6 expression level typically decreases while the MYH7 level increases. However, in the present study, the addition of T/D decreased the MYH7 level while increasing the MYH6 level. Please comment on this difference.

Response: In humans, it is true that Myh6 level decreases with age and Myh7 increases with age. In rodents, it is the opposite: Myh6 increases with age and Myh7 decreases with age. 

In the text, lines 193-194, we put one sentence to highlight the difference between human and rodents.

"Myosin heavy chain 6 (Myh6) and Myh7 are two cardiac muscle genes abundantly expressed in adult and fetal murine CMs, respectively [21]. In adult human heart, the dominant myosin heavy chain isoform is MYH7 [22]." 

The TNNI3 level is also an important maturation marker, which could be up-regulated during the maturation process. The reviewer is curious whether there is a change in the present study. Please analyze the expression level (immunostaining) in Figure 2A and Figure 3A.

Response: We analyzed TNNI3 intensity. The new data (Suppl. Fig. 2E) were included in the revision. 

Lines 185-189:

"During CM maturation, immature slow skeletal slow skeletal troponin I is gradually replaced by mature cardiac troponin I [20]. We then assessed TNNI3 expression levels by measuring its fluorescence intensity. The TNNI3 fluorescence intensity per cell was similar between control and Dex treated NRVMs, and T3 or T3/D treated NRVMs showed much stronger TNNI3 fluorescence intensity than the control and Dex treated CMs (Suppl. Fig. 2E).

In the statistical method section, please specify which data belong to normally distributed data and which are non-parametric data.

Response: We specified the data categories in the statistic section.

"Normally distributed data values, such as mRNA and protein levels, were expressed as mean ± SD, and analyzed with student’s t-test (two groups) or one-way ANOVA followed by Tukey’s post hoc test (more than two groups). Non-parametric data, such as cell size, myofibril thickness/length, were analyzed using Mann-Whitney test (two groups) or Kruskall-Wallis test followed by Dunn's multiple comparisons (more than two groups).". 

Please specify the full names of FCCP and R/A in the OCR data figure legend for better reader understanding. Additionally, the OCR data show that T/D has a weaker effect than T3 alone. Please discuss the possible reasons for this.

Response: The full name were provided. We put two sentences to discuss this observation: Line 378-383:

"Our OCR data suggest that Dex diminishes T3's metabolic effects. Since Dex may counteract T3 action either through enhancing T3 to metabolic inactive reverse T3 (rT3) conversion [37] or by decreasing Thyroid hormone receptor expression [38], we reasoned that T3's strong metabolic effects can be dampened by Dex, so that the T3/Dex treated CMs have a better balanced metabolism program than the T3-treated CMs"

Reviewer 3 Report

Comments and Suggestions for Authors

In this study, Farley and colleagues investigate the role of VGLL4 in cardiomyocyte (CM) maturational hypertrophic growth. Through carefully planned in vitro and in vivo experiments with appropriate controls, the authors demonstrate a convincing mechanism by which VGLL4 promotes TEAD degradation and disrupts the YAP/TAZ-TEAD complex to suppress maturational hypertrophic growth. The presented findings and proposed mechanism leave an impactful contribution to the field of cardiac development. Taken together, this manuscript should be accepted for publication following minor revisions:

Major Comment:

1.     The introduction can be strengthened by contextualizing how the recent boom in employing hiPSC-CMs to study cardiac development and/or disease modeling makes this study timely and impactful. This was briefly discussed in the first paragraph of the discussion section, and should be moved to the intro.

Minor Comment:

2.     Figures: Figure 1 has incorrect panel numberings and is missing E-F. Figure 2 legend should be labeled above panels B-E, similar to how it was done for Figures 3-8.

3.     Typos:

- Line 185: missing “of” in “… expression of the aforementioned …”

- Line 199: metabolic is misspelled

- Figure 2K: Baseline is misspelled

Author Response

Reviewer 3: 

In this study, Farley and colleagues investigate the role of VGLL4 in cardiomyocyte (CM) maturational hypertrophic growth. Through carefully planned in vitro and in vivo experiments with appropriate controls, the authors demonstrate a convincing mechanism by which VGLL4 promotes TEAD degradation and disrupts the YAP/TAZ-TEAD complex to suppress maturational hypertrophic growth. The presented findings and proposed mechanism leave an impactful contribution to the field of cardiac development. Taken together, this manuscript should be accepted for publication following minor revisions:

Major Comment:

1.     The introduction can be strengthened by contextualizing how the recent boom in employing hiPSC-CMs to study cardiac development and/or disease modeling makes this study timely and impactful. This was briefly discussed in the first paragraph of the discussion section, and should be moved to the intro.

Response: We thank the reviewer for this nice suggestion. To highlight the rational of the current study, we put two sentences in the introduction section. 

"The mechanisms that regulate and implement maturation are being robustly investigated due to the recent advance in induced pluripotent stem cell (iPSC)-based cardiac regeneration medicine, because a major barrier hampering the progress of this field is the limited maturity of CMs that can be generated from iPS. One of the fruitful strategy to understand CM maturation is to study the molecular signaling pathways controlling the development of naturally occurred mammalian CMs, such as neonatal rodent CMs, and the new knowledge originated from these studies will facilitate the effort of generating iPSC-derived mature CMs" 

Minor Comment:

2.     Figures: Figure 1 has incorrect panel numberings and is missing E-F. Figure 2 legend should be labeled above panels B-E, similar to how it was done for Figures 3-8.

Response: These errors have been fixed. 

3.     Typos:

- Line 185: missing “of” in “… expression of the aforementioned …”

- Line 199: metabolic is misspelled

- Figure 2K: Baseline is misspelled

Response: These typos have been corrected. 
